# *Coronin 1C*, Regulated by Multiple microRNAs, Facilitates Cancer Cell Aggressiveness in Pancreatic Ductal Adenocarcinoma

**DOI:** 10.3390/genes14050995

**Published:** 2023-04-27

**Authors:** Kosuke Fukuda, Naohiko Seki, Ryutaro Yasudome, Reiko Mitsueda, Shunichi Asai, Mayuko Kato, Tetsuya Idichi, Hiroshi Kurahara, Takao Ohtsuka

**Affiliations:** 1Department of Digestive Surgery, Breast and Thyroid Surgery, Graduate School of Medical and Dental Sciences, Kagoshima University, Kagoshima 890-8520, Japan; k3624054@kadai.jp (K.F.); k7682205@kadai.jp (R.Y.); k7854142@kadai.jp (R.M.); k3352693@kadai.jp (T.I.); k7313641@kadai.jp (H.K.); takao-o@kufm.kagoshima-u.ac.jp (T.O.); 2Department of Functional Genomics, Graduate School of Medicine, Chiba University, Chiba 260-8670, Japan; cada5015@chiba-u.jp (S.A.); mayukokato@chiba-u.jp (M.K.)

**Keywords:** pancreatic ductal adenocarcinoma, coronin-family, *CORO1C*, microRNA, *miR-26a-5p*, *miR-29c-3p*

## Abstract

Coronin proteins are actin-related proteins containing WD repeat domains encoded by seven genes (*CORO1A*, *CORO1B*, *CORO1C*, *CORO2A*, *CORO2B*, *CORO6*, and *CORO7*) in the human genome. Analysis of large cohort data from The Cancer Genome Atlas revealed that expression of *CORO1A*, *CORO1B*, *CORO1C*, *CORO2A*, and *CORO7* was significantly upregulated in pancreatic ductal adenocarcinoma (PDAC) tissues (*p* < 0.05). Moreover, high expression of *CORO1C* and *CORO2A* significantly predicted the 5 year survival rate of patients with PDAC (*p* = 0.0071 and *p* = 0.0389, respectively). In this study, we focused on *CORO1C* and investigated its functional significance and epigenetic regulation in PDAC cells. Knockdown assays using siRNAs targeting *CORO1C* were performed in PDAC cells. Aggressive cancer cell phenotypes, especially cancer cell migration and invasion, were inhibited by *CORO1C* knockdown. The involvement of microRNAs (miRNAs) is a molecular mechanism underlying the aberrant expression of cancer-related genes in cancer cells. Our *in silico* analysis revealed that five miRNAs (*miR-26a-5p*, *miR-29c-3p*, *miR-130b-5p*, *miR-148a-5p*, and *miR-217*) are putative candidate miRNAs regulating *CORO1C* expression in PDAC cells. Importantly, all five miRNAs exhibited tumor-suppressive functions and four miRNAs except *miR-130b-5p* negatively regulated *CORO1C* expression in PDAC cells. *CORO1C* and its downstream signaling molecules are potential therapeutic targets in PDAC.

## 1. Introduction

Pancreatic ductal adenocarcinoma (PDAC) is one of the deadliest human cancers, with a 5 year survival rate of approximately 10%, regardless of the stage [1,2,3]. In Japan, the annual number of deaths from PDAC is approximately 38,000 among both men and women, ranking as the fourth deadliest cancer after lung cancer, colorectal cancer, and gastric cancer [4].

Long-term survival is expected in patients with early-stage PDAC and tumors that can be surgically resected. However, only about 20% of all pancreatic cancers can be surgically resected. Furthermore, approximately 30% of patients already have locally advanced PDAC at diagnosis, and 50% of patients have metastatic disease [2,3,5]. Chemotherapy regimens, e.g., FOLFIRINOX (5-fluorouracil, folinic acid, irinotecan, and oxaliplatin), gemcitabine-containing combination treatments, and gemcitabine treatments, are indicated for patients who are ineligible for surgery [2,5,6]. Despite these treatments, the survival time of patients with advanced-stage PDAC is less than 1 year [2,3,6].

Previous studies have shown that four major genetic mutations (*KRAS*, *TP53*, *CDKN2A*, and *SMAD4*) are closely involved in the oncogenic process of PDAC [6,7,8]. Recent advances in sequencing technology have enabled whole-genome mutation analyses in individual cancer cells, thereby revealing genomic differences among individual cancer cells. The molecular classification of PDAC (the Know Your Tumor program) has revealed that approximately 25% of PDAC patients have actionable molecular alterations [9]. Recently, several therapeutic strategies that target actionable driver gene mutations in cancer cells have been developed. For example, PARP inhibitors target PDAC cells with *BRCA1* or *BRCA2* mutations, and immune checkpoint inhibitors target cells with mismatch repair deficiencies [3,5,7,10].

PDAC patients frequently develop recurrence or metastasis even after curative resection [11]. To improve the prognosis of PDAC, it is essential to clarify the molecular mechanisms of recurrence and metastasis of PDAC cells. To date, no driver molecules causing recurrence or metastasis have been identified. A unique feature of PDAC is the abundant fibrous stroma composed of extracellular matrix (ECM) proteins surrounding the cancer cells [12,13]. Cell invasion through the ECM of cancer cells is the first step in metastasis. The formation of various structures, such as invadopodia and pseudopodia, is required for PDAC cells to degrade and migrate through the ECM. These structural changes require actin assembly regulated by specialized actin nucleation factors. Reorganization of the actin cytoskeleton is essential for invasive cell migration and is initiated by various actin nucleation factors [14]. Actin nucleators are now emerging as promising targets to control cancer cell metastasis.

In this study, we focused on coronin proteins, which bind to filamentous actin and the Arp2/3 complex and contribute to modulating actin dynamics [15]. In humans, the coronin family consists of seven genes (*CORO1A*, *CORO1B*, *CORO1C*, *CORO2A*, *CORO2B*, *CORO6*, and *CORO7*), all of which possess a conserved basic N-terminal motif and several WD repeats clustered in the core domains [16]. Analysis of The Cancer Genome Atlas (TCGA) data revealed that overexpression of *CORO1C* and *CORO2A* is closely involved in PDAC molecular pathogenesis. Functional assays of *CORO1C* revealed that aberrant expression of *CORO1C* facilitates PDAC cell migration and invasion abilities.

We analyzed the aberrant expression of *CORO1C* in PDAC cells from an epigenomic viewpoint. A vast number of studies have shown that microRNAs (miRNAs) negatively regulate gene expression [17,18]. Downregulation of tumor-suppressive miRNAs induces overexpression of cancer-promoting genes in cancer cells [19,20]. Our analysis revealed that four miRNAs (*miR-26a-5p*, *miR-29c-3p*, *miR-148a-5p*, and *miR-217*) regulated *CORO1C* expression in PDAC cells. All these miRNAs acted as tumor-suppressive miRNAs in PDAC cells.

## 2. Materials and Methods

### 2.1. Analysis of the Expression and Clinical Significance of Coronin Genes in PDAC

The expression levels of the coronin genes *CORO1A*, *CORO1B*, *CORO1C*, *CORO2A*, *CORO1B*, *CORO6*, and *CORO7* in PDAC clinical specimens were evaluated using the GEPIA2 platform (http://gepia2.cancer-pku.cn/#index; accessed on 31 May 2022). For the Kaplan–Meier plots and log-rank test, we used the OncoLnc database (http://www.oncolnc.org, accessed on 13 June 2022).

### 2.2. Gene Set Enrichment Analysis (GSEA)

GSEA was performed to investigate the CORO1C-mediated molecular pathways. The TCGA–PDAC data were divided into high- and low-expression groups according to the Z-score of the *CORO1C* expression level. The expression levels of each gene were compared in the high- and low-*CORO1C* expression groups, and the genes were ranked according to the log_2_ ratio. We uploaded the resultant ranked gene lists into GSEA software [21,22] and applied the Hallmark gene set in the Molecular Signatures Database [23].

### 2.3. Human PDAC Cell Lines

Two PDAC cell lines, PANC-1 and SW1990, were used in this study. The cell lines were purchased from the American Type Culture Collection (Manassas, VA, USA) and the RIKEN Cell Bank (Tsukuba, Japan), respectively.

### 2.4. Transfection of Small Interfering RNAs (siRNAs) and miRNAs into PDAC Cells and Quantitative Reverse-Transcription PCR (qRT-PCR)

Transfection of siRNAs and miRNAs into PDAC cell lines was performed using Lipofectamine RNAiMAX reagent (Invitrogen, Carlsbad, CA, USA) according to our previous studies [24,25,26]. In this study, we define “mock” and “control” as follows: mock: transfection reagent was added to the medium; control: scrambled RNA different from the siRNA or miRNA sequence was added to the medium. The reagents used in this study are listed in Appendix A. We performed qRT-PCR using the StepOnePlus™ Real-Time PCR System (Applied Biosystems, Waltham, MA, USA), and gene expression levels were normalized to those of *GUSB* as the internal control. The sequences of the SYBR Green primers are listed in Appendix A. The procedures for qRT-PCR assays in PDAC cells have been described previously [24,25,26].

### 2.5. Functional Assays (Cell Proliferation, Migration, and Invasion) in PDAC Cells

The procedures for the functional assays (cell proliferation, migration, and invasion assays) in PDAC cells have been described previously [24,25,26]. Cell proliferation was examined by the XTT assay (Sigma–Aldrich, St. Louis, MO, USA) at 72 h after miRNA or siRNA transfection. Cell migration and invasion assays were conducted using the BioCoatTM cell culture chamber and the BioCoat Matrigel Invasion Chamber, both from Corning (Corning, NY, USA). Forty-eight hours after transfection, the cells at the bottom of the chamber were counted and analyzed.

### 2.6. Identification of the miRNAs Regulating CORO1C Expression in PDAC Cells

The Target Scan Human 8.0 database (http://www.targetscan.org/vert_80, accessed on 21 September 2022) was used to select miRNAs that have putative binding sites in the 3′UTR of *CORO1C*. The miRNA expression signature of PDAC was used to screen miRNAs [27]. Expression of miRNAs in PDAC clinical specimens was assessed using the GEO database (GSE24279 and GSE71533).

### 2.7. Western Blotting

The Western blotting procedure has been described previously [24,25,26]. The anti-CORO1C antibody (Proteintech Group, Inc., Rosemont, IL, USA) was diluted 1:2500, and the anti-GAPDH antibody (Wako, Osaka, Japan), used as the internal control, was diluted 1:1600. The antibodies used are listed in Appendix A.

### 2.8. Statistical Analysis

JMP Pro 15 software (SAS Institute Inc., Cary, NC, USA) was used for the statistical analyses. Differences between the two groups were evaluated using Welch’s *t*-test, and differences among multiple groups were evaluated using Dunnett’s test. To analyze prognosis based on *CORO1C* expression, the patients were divided into two groups according to the expression level of *CORO1C*, and differences in survival were evaluated. A *p*-value less than 0.05 was considered statistically significant.

## 3. Results

### 3.1. Expression and Clinical Significance of Coronin Family Members in Patients with PDAC

Based on the TCGA and GEPIA2 databases, the expression levels of all members of the coronin family (*CORO1A*, *CORO1B*, *CORO1C*, *CORO2A*, *CORO2B*, *CORO6*, and *CORO7*) were analyzed. Among the coronin genes, the expression levels of five (*CORO1A*, *CORO1B*, *CORO1C*, *CORO2A*, and *CORO7*) were significantly upregulated in PDAC tissues (*n* = 179) compared with normal pancreatic tissues (*n* = 179) (*p* < 0.05, Figure 1A). Two genes (*CORO2B* and *CORO6*) did not show significant expression in PDAC tissues (Figure 1A).

Next, we investigated whether high expression of these genes affected patient prognosis using TCGA PDAC data. The prognosis was significantly worse in patients with high expression of *CORO1C* and *CORO2A* versus low expression of these genes (*p* = 0.0071 and *p* = 0.0389, respectively) (Figure 1B). Notably, the prognosis was significantly worse in patients with low versus high expression of *CORO2B* (*p* = 0.0144, Figure 1B). As a result of the *in silico* analysis, two genes (*CORO1C* and *CORO2A*) were selected as candidate cancer-promoting genes that affect the prognosis of PDAC patients. Molecular analysis of these two genes is essential for elucidating the malignant transformation of PDAC cells. By comparing the significant difference in the 5 year survival rate, this study focused on *CORO1C* and performed functional analysis.

### 3.2. Molecular Pathways Associated with High CORO1C Expression in PDAC Cells

Using TCGA PDAC data, we performed GSEA to identify the molecular pathways activated in patients with high *CORO1C* expression. We found that “epithelial–mesenchymal transition”, “inflammatory response”, and “KRAS signaling” were pathways enriched in the high *CORO1C* expression group (Figure 2). Activation of these pathways could potentially accelerate the malignant transformation of PDAC cells.

### 3.3. Effects of CORO1C Knockdown on PDAC Cell Proliferation, Migration, and Invasion

To assess the oncogenic function of *CORO1C* in PDAC cells, we performed knockdown assays using siRNAs. The inhibitory effect of two different siRNAs targeting *CORO1C* (si*CORO1C*-1 and si*CORO1C*-2) on *CORO1C* expression was examined. The *CORO1C* mRNA and protein levels were effectively suppressed by transfection of both siRNAs into two PDAC cell lines, PANC-1 and SW1990 (Appendix A).

Then, functional assays using these siRNAs were performed. The knockdown of *CORO1C* had little effect on cell proliferation in PANC-1 and SW1990 cells (Figure 3A).

Cell migration and invasion were significantly suppressed after transfection of the *CORO1C* siRNAs into PANC-1 and SW1990 cells (Figure 3B,C). Typical images of cells during the migration and invasion assays after si*CORO1C* transfection are shown in Figure 3D,E.

The antitumor effects (cell proliferation and migration) of the two siRNAs, si*CORO1C*-1 and si*CORO1C*-2, differed (Figure 3A,B). Under the analysis conditions of this study, the antitumor effects of si*CORO1C*-1 were insufficient compared to si*CORO1C*-2 in SW1990 cells.

### 3.4. Identification of miRNAs That Regulate CORO1C Expression in PDAC Cells

A vast number of studies have revealed that gene expression is regulated by various functional RNAs, e.g., lncRNAs, miRNAs, and circRNAs. Among them, it has been clarified that dysregulation of miRNAs causes aberrant expression of cancer-related genes and is closely involved in the malignant transformation of cancer cells. Therefore, we hypothesized that downregulation of some miRNAs might cause overexpression of *CORO1C* in PDAC cells. We searched for miRNAs that negatively regulate *CORO1C* expression in PDAC cells.

The strategy for identifying miRNAs that regulate *CORO1C* expression is shown in Figure 4. The combination of TargetScan data (release 8.0) and our miRNA signature of PDAC revealed 35 candidate miRNAs that regulate *CORO1C* expression in PDAC cells (Table 1). The expression levels of the 35 miRNAs were evaluated in GEO datasets (GSE 24279 and GSE 71533). Five miRNAs (*miR-26a-5p*, *miR-29c-3p*, *miR-130b-5p*, *miR-148a-5p*, and *miR-217*) were commonly downregulated in two datasets. We confirmed that expression levels of five miRNAs were downregulated in PDAC tissues (*n* = 136) compared with normal tissues (*n* = 22) by using GSE 24279 data (Figure 5A). Analysis using another dataset (GSE71533) also confirmed that the expression levels of five miRNAs were suppressed in PDAC tissues (*n* = 36) compared with normal tissues (*n* = 16) (Figure 5B).

The correlations of expression levels between five miRNAs (*miR-26a-5p*, *miR-29c-3p*, *miR-130b-5p*, *miR-148a-5p,* and *miR-217*) and *CORO1C* were evaluated using TCGA-PDAC data (Appendix A). A Spearman’s correlation coefficient rank test indicated that a negative correlation was detected in the expression levels of *miR-29c-3p* and *CORO1C* in PDAC clinical specimens (*p* < 0.05, *r* = −0.3666). No inverse correlation was observed between the expression levels of other miRNAs and *CORO1C*.

Next, we examined whether these miRNAs control the expression of *CORO1C* in PDAC cells. *CORO1C* mRNA expression after miRNA transfection. *CORO1C* mRNA expression was reduced at 48 h after transfection of *miR-26a-5p*, *miR-29c-3p*, *miR-148a-5p*, or *miR-217* in PDAC cell lines (PANC-1 and SW1990). On the other hand, transfection with *miR-130b-5p* did not show remarkable regulation of *CORO1C* expression in PDAC cells (Figure 6A). CORO1C protein expression was reduced at 72 h after transfection of *miR-26a-5p*, *miR-29c-3p*, *miR-148a-5p*, or *miR-217* in PDAC cells (Figure 6B). The intensity of the bands for the Western blotting was analyzed using ImageJ software. The results of the ImageJ analyses are shown in Appendix A.

### 3.5. Tumor-Suppressive Function of miR-26a-5p and miR-29c-3p in PDAC Cells

Our previous studies revealed that *miR-130b-5p*, *miR-148a-5p*, and *miR-217* act as tumor-suppressive miRNAs in PDAC cells by targeting several oncogenes [28,29,30]. Therefore, in this study, we analyzed whether the two miRNAs, *miR-26a-5p* and *miR-29c-3p*, have tumor-suppressive functions in PDAC cells.

The tumor-suppressive activities of *miR-26a-5p* and *miR-29c-3p* were assessed by ectopic expression of mature miRNAs in PANC-1 and SW1990 cells. The inhibitory effect of *miR-26a-5p* on cell proliferation was observed in SW1990 cells but not in PANC-1 cells (Figure 7A). Cancer cell invasion and migration abilities were markedly suppressed by ectopic expression of *miR-26a-5p* in PDAC cells (Figure 7B,C). Typical images of Figure 7B (cell migration) and Figure 7C (cell invasion) are shown in Figure 7D,E, respectively.

Similar to *miR-26a-5p*, ectopic expression of *miR-29c-3p* attenuated the malignant phenotype, i.e., cell proliferation, invasion, and migration abilities, of PDAC cells (Figure 8A–C). Typical images of Figure 8B (cell migration) and Figure 8C (cell invasion) are shown in Figure 8D,E, respectively.

## 4. Discussion

The main cause of death in cancer patients is malignant transformation and metastasis of cancer cells rather than the surgically removable primary tumor. To improve the prognosis of PDAC patients, it is essential to clarify the molecular mechanism of pancreatic cancer metastasis.

In this study, we analyzed the coronin family of WD-repeat actin-binding proteins. Previous studies showed that coronin proteins regulate actin-dependent processes via F-actin assembly [15,16]. TCGA analysis revealed that overexpression of *CORO1C* is closely involved in the molecular pathogenesis of PDAC. Aberrant expression of *CORO1C* has been reported in several types of human cancers, e.g., glioblastoma, hepatocellular cancer, breast cancer, non-small cell lung cancer, gastric cancer, and colorectal cancer [31,32,33,34,35,36,37,38]. In addition, high expression of *CORO1C* is closely associated with a worse prognosis and aggressive pathological parameters in hepatocellular carcinoma, gastric cancer, and colorectal cancer [32,37,38]. Furthermore, we investigated the CORO1C-mediated molecular pathways in PDAC. GSEA analysis revealed that “epithelial–mesenchymal transition (EMT)” was the most enriched pathway in the *CORO1C* high-expression group. EMT has widely been accepted as a critical step in metastatic dissemination in PDAC cells, and various functional RNAs are closely involved in this process [39,40]. However, EMT-independent pathways have also been reported to be involved in distant metastases of PDAC [41]. Exploring the molecular pathways mediated by *CORO1C* will provide hints for elucidating the invasion and metastasis mechanisms of PDAC cells.

Previous functional analyses of *CORO1C* have shown that it functions as a cancer-promoting gene in several cancer cells [32,33,34,37,38]. In breast cancer cells, *CORO1C* was found to be involved in invadopodia formation and MT1-MMP surface trafficking and to promote invasiveness [35]. In gastric cancer cells, *CORO1C* expression facilitated cancer cell aggressiveness (e.g., cell viability, colony formation, and metastasis) by positively regulating cyclin D1 and vimentin expression [37]. In colorectal cancer cells, *CORO1C* interacted with trophoblast cell surface protein 2, and *CORO1C* overexpression enhanced cancer phenotypes via the PI3K/AKT signaling pathway [38]. The multifaceted functions of *CORO1C* enable it to activate various molecular pathways, which may accelerate the malignant transformation of cancer cells. Aberrant activation of the PI3K/AKT/mTOR pathway strongly contributed to the malignant transformation of various cancers, including PDAC [42,43]. Therefore, various inhibitors have been developed to block this oncogenic pathway [44]. Currently, PI3K/AKT/mTOR inhibitors are being tested in vitro and in vivo with promising results in PDAC patients [45].

Furthermore, in this study, we investigated the molecular mechanism of aberrant expression of *CORO1C* in PDAC cells from an epigenomic viewpoint. miRNAs are small RNA molecules that act as fine tuners of gene expression, and aberrant expression of miRNAs is involved in malignant transformation, drug resistance, and metastasis of cancer cells [19]. Our research group revealed the miRNA expression signature of PDAC based on RNA sequencing, and we are continuing to identify tumor-suppressive miRNAs and their oncogenic targets [24,27,28,29,30,46,47,48,49,50].

In this study, our analysis revealed that four miRNAs (*miR-26a-5p*, *miR-29c-3p*, *miR-148a-5p*, and *miR-217*) negatively regulate *CORO1C* expression in PDAC cells. Previous reports showed that *miR-26a* expression was reduced in PDAC tissues and that overexpression of *miR-26a* attenuated PDAC cell proliferation [51,52]. Another study showed that expression of *miR-26a-5p* blocked cancer cell malignant phenotypes by targeting the aryl hydrocarbon receptor nuclear translocator like 2, which plays crucial roles in the oncogenesis of multiple cancers [53].

Regarding *miR-29c*, overexpression of *miR-29c* in PDAC cells was found to enhance gemcitabine sensitivity by inhibiting autophagy activation [54]. Downregulation of *miR-29c* was detected in clinical PDAC tissues, and this miRNA showed an antitumor function by inhibiting integrin subunit β1 [55]. A recent study showed that *miR-29c* targets *MAPK1* to suppress activation of the ERK/MAPK pathway [56].

According to the traditional concept of miRNA biogenesis, the miRNA guide strands derived from pre-miRNAs are incorporated into the RNA-induced silencing complex and suppress the expression of their target genes. In contrast, the miRNA passenger strands are degraded in the cytoplasm and appear to have no function [57]. Our recent studies demonstrated that some passenger strands of miRNAs act as tumor suppressors by targeting several cancer-related genes and oncogenic pathways in several types of cancer, including PDAC [26,28,29,58,59,60]. We also demonstrated that overexpression of the passenger strands *miR-130b-5p* and *miR-148a-5p* attenuated PDAC cell malignant phenotypes by regulating several genes involved in PDAC molecular pathogenesis [28,29].

Numerous studies have reported that *miR-217* is downregulated in multiple types of cancers and that expression of *miR-217* blocks cancer cell proliferation, metastasis, epithelial–mesenchymal transition, and drug resistance [61]. Several *miR-217*-regulated genes are strongly associated with cancer diagnosis and prognosis [58]. Our miRNA signature in PDAC indicated that *miR-217* was one of the most downregulated miRNAs, and its expression blocked PDAC cell aggressiveness by targeting the actin-binding protein Anillin [30].

Thus, it is a very interesting finding that *CORO1C* expression is regulated by multiple tumor-suppressive miRNAs. Downregulation of these tumor-suppressive miRNAs may be a key factor in the molecular mechanism leading to overexpression of *CORO1C* in PDAC cells.

## 5. Conclusions

Analysis of TCGA and GEPIA2 data revealed that expression of *CORO1C* is closely involved in PDAC molecular pathogenesis. Functional assays showed that overexpression of *CORO1C* facilitated PDAC cell aggressiveness. Four miRNAs (*miR-26a-5p*, *miR-29c-3p*, *miR-148a-5p*, and *miR-217*) negatively regulated *CORO1C* expression in PDAC cells. These miRNAs were downregulated in PDAC tissues, and they acted as tumor-suppressive miRNAs in PDAC cells. In summary, *CORO1C* may be a therapeutic target for PDAC, and control of this molecule will lead to improved prognosis in patients with PDAC.

## Figures and Tables

**Figure 1 genes-14-00995-f001:**
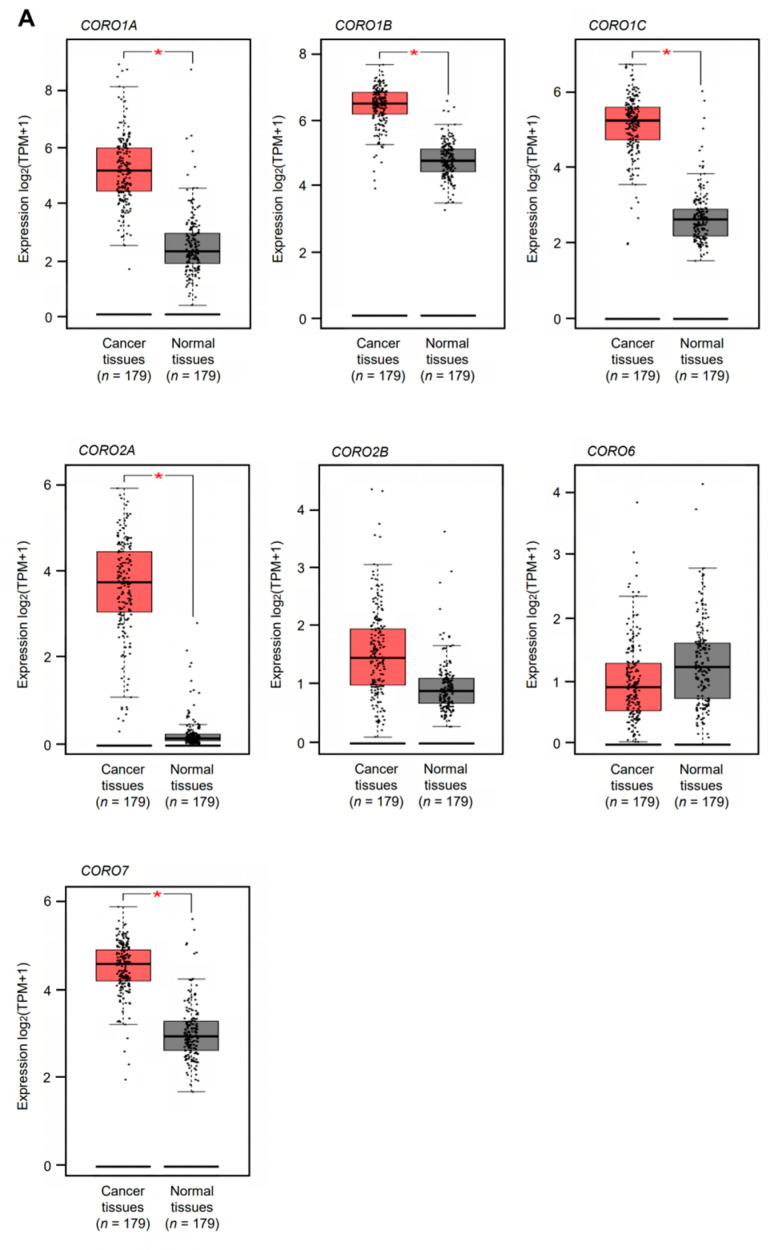
Expression and clinical significance of all coronin family members based on TCGA analysis. (**A**) Expression levels of *CORO1A*, *CORO1B*, *CORO1C*, *CORO2A*, *CORO2B*, *CORO6*, and *CORO7* in PDAC tissues. A total of 179 PDAC tissues and 179 normal pancreatic tissues were analyzed (* *p* < 0.05). (**B**) Kaplan–Meier survival analysis of patients with PDAC using the TCGA PDAC dataset. The patients were divided into high- and low-expression groups according to miRNA expression (based on a median expression level). The red line represents the high-expression group, and the blue line represents the low-expression group.

**Figure 2 genes-14-00995-f002:**
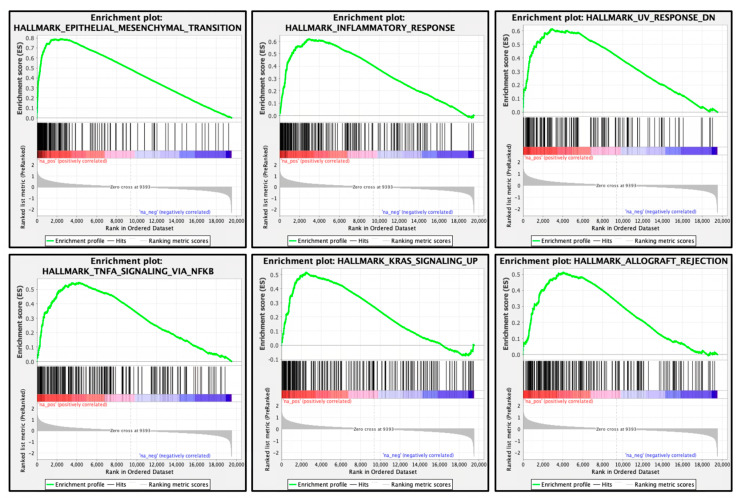
CORO1C-mediated pathways identified by gene set enrichment analysis. The top six enrichment plots in the high *CORO1C* expression group.

**Figure 3 genes-14-00995-f003:**
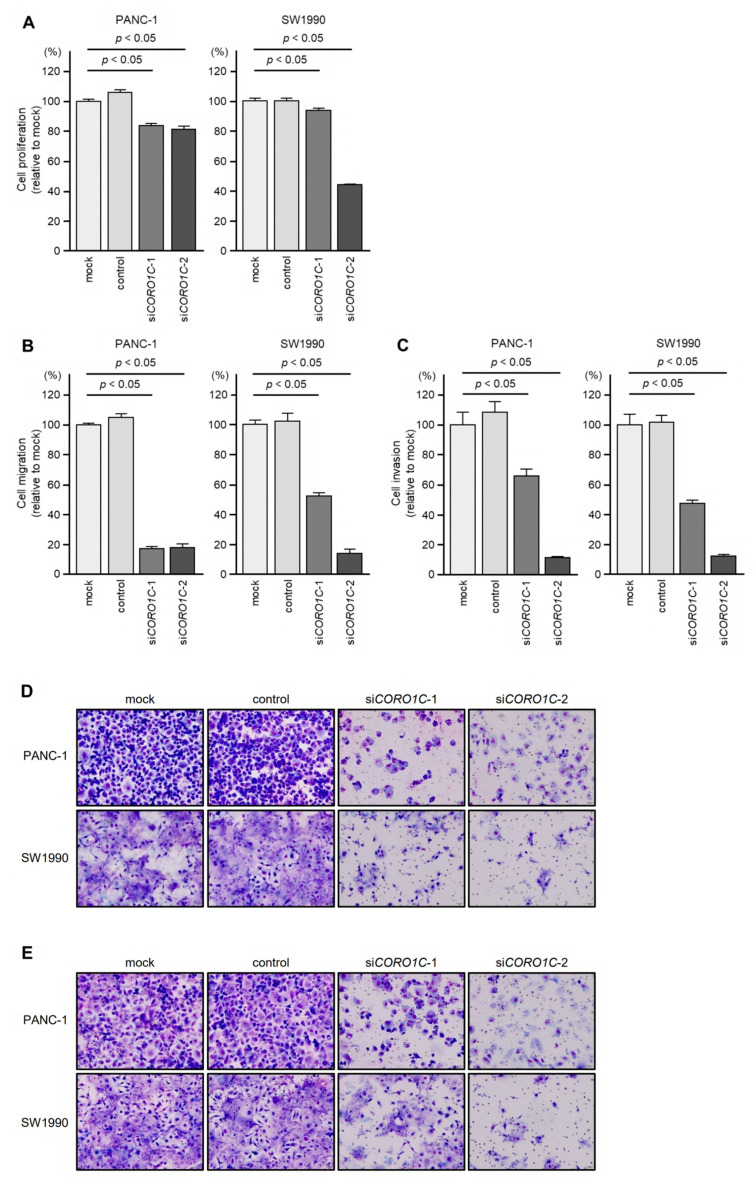
Functional assays of cell proliferation, migration, and invasion following transient transfection of siRNAs (si*CORO1C-1* and si*CORO1C*-2) in two PDAC cell lines (PANC-1 and SW1990). (**A**) Cell proliferation is assessed by the XTT assay 72 h after siRNA transfection. (**B**) Cell migration is assessed by a membrane culture system 48 h after seeding siRNA-transfected cells into the chambers. (**C**) Cell invasion is assessed by Matrigel invasion assays 48 h after seeding siRNA-transfected cells into the chambers. (**D**,**E**) Photographs of typical results from the migration (**D**) and invasion (**E**) assays.

**Figure 4 genes-14-00995-f004:**
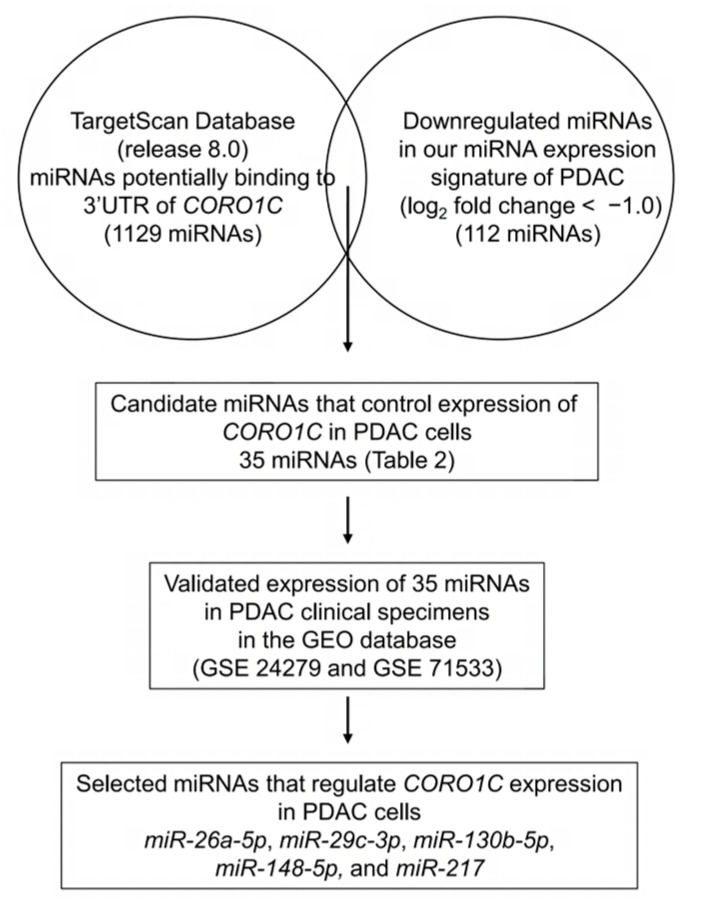
Flowchart of the strategy used to identify candidate miRNAs controlling *CORO1C* expression in PDAC cells.

**Figure 5 genes-14-00995-f005:**
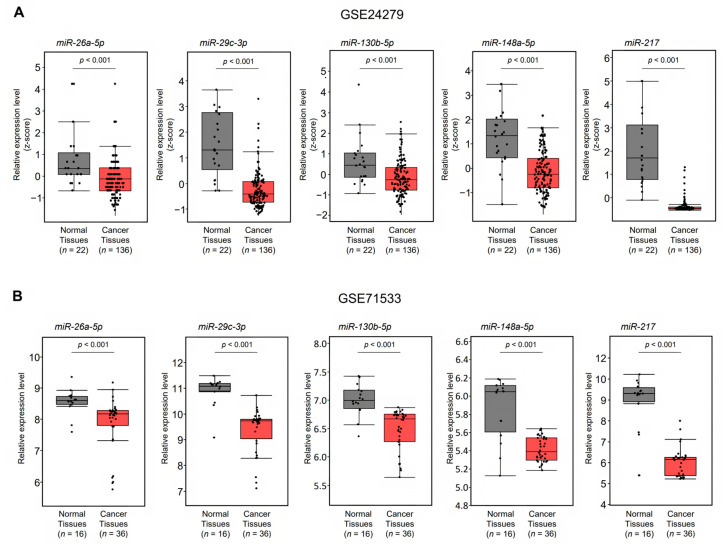
Expression of *miR-26a-5p*, *miR-29c-3p*, *miR-130b-5p*, *miR-148a-5p*, and *miR-217* in PDAC tissues. (**A**,**B**) The expression levels of five miRNAs were evaluated using GEO databases GSE24279 (**A**) and GSE71533 (**B**).

**Figure 6 genes-14-00995-f006:**
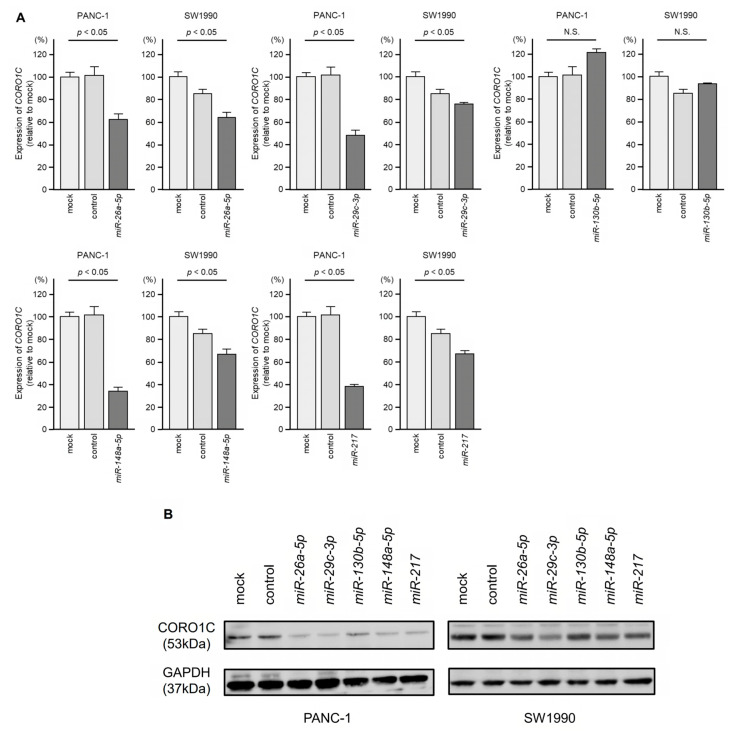
Regulation of *CORO1C* expression by five miRNAs (*miR-26a-5p*, *miR-29c-3p*, *miR-130b-5p*, *miR-148a-5p*, and *miR-217*) in PDAC cells. (**A**) *CORO1C* mRNA expression after five miRNA transfections. (**B**) CORO1C protein expression according to Western blotting after five miRNA transfections.

**Figure 7 genes-14-00995-f007:**
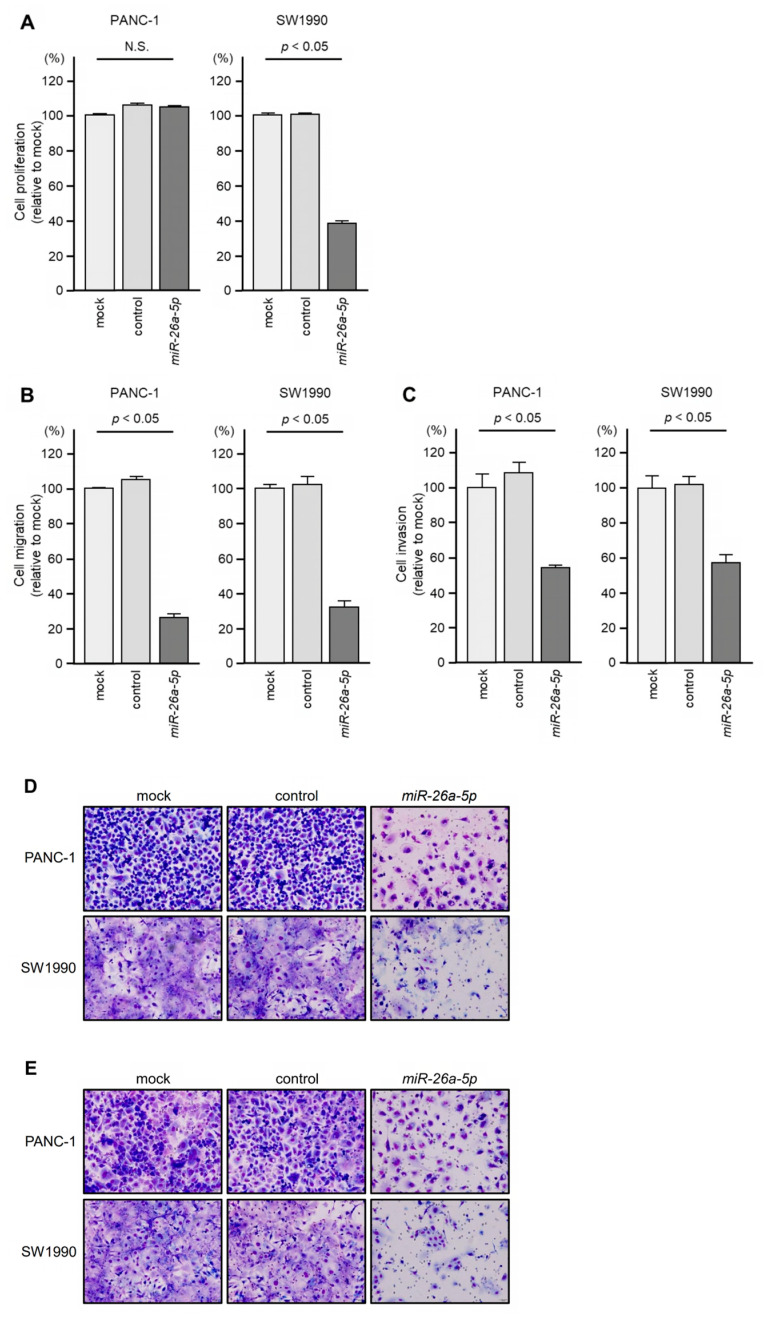
Tumor-suppressive function of *miR-26a-5p* in PDAC cells. (**A**) Cell proliferation is assessed by the XTT assay 72 h after transfection of mature miRNAs. (**B**) Cell migration is assessed by a membrane culture system 48 h after seeding miRNA-transfected cells into the chambers. (**C**) Cell invasion is assessed by Matrigel invasion assays 48 h after seeding miRNA-transfected cells into the chambers. (**D**,**E**) Photographs of typical results from the migration (**D**) and invasion (**E**) assays.

**Figure 8 genes-14-00995-f008:**
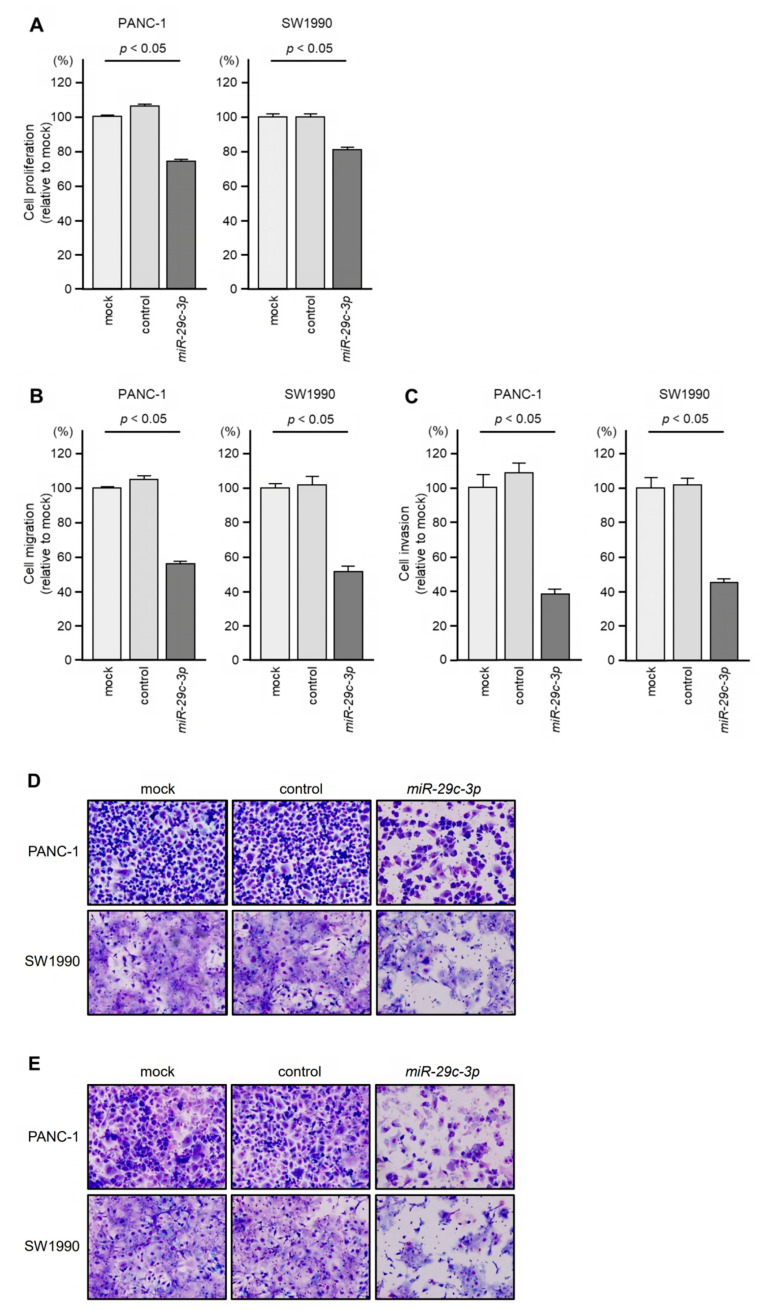
Tumor-suppressive function of *miR-29c-3p* in PDAC cells. (**A**) Cell proliferation is assessed by the XTT assay 72 h after transfection of mature miRNAs. (**B**) Cell migration is assessed by a membrane culture system 48 h after seeding miRNA-transfected cells into the chambers. (**C**) Cell invasion is assessed by Matrigel invasion assays 48 h after seeding miRNA-transfected cells into the chambers. (**D**,**E**) Photographs of typical results from the migration (**D**) and invasion (**E**) assays.

**Table 1 genes-14-00995-t001:** Candidate miRNAs that regulate *CORO1C* expression in PDAC cells.

miRNA	miRbase Accession	Chromosome	FC (log_2_)	*p* Value	FDR
*hsa-miR-217*	*MIMAT0000274*	2	−3.1333	0.003394591	0.460235049
*hsa-miR-216a-3p*	*MIMAT0022844*	2	−2.5792	0.000762794	0.172184571
*hsa-miR-129-1-3p*	*MIMAT0004548*	7	−2.4045	0.001131503	0.208196466
*hsa-miR-148a-5p*	*MIMAT0004549*	7	−2.3043	0.000857068	0.172184571
*hsa-miR-211-5p*	*MIMAT0000268*	15	−2.1027	0.008394937	0.554496371
*hsa-miR-129-2-3p*	*MIMAT0004605*	11	−2.0092	0.008992112	0.564967844
*hsa-miR-2114-3p*	*MIMAT0011157*	X	−1.9620	0.025664949	0.920037824
*hsa-miR-7-2-3p*	*MIMAT0004554*	15	−1.9294	0.118440035	1
*hsa-miR-4780*	*MIMAT0019939*	2	−1.7904	0.063787284	1
*hsa-miR-129-5p*	*MIMAT0000242*	7	−1.6670	0.063243326	1
*hsa-miR-204-5p*	*MIMAT0000265*	9	−1.6320	0.028968095	0.969114438
*hsa-miR-130b-5p*	*MIMAT0004680*	22	−1.5926	0.006911516	0.502959718
*hsa-miR-19a-3p*	*MIMAT0000073*	13	−1.3565	0.072163518	1
*hsa-miR-7855-5p*	*MIMAT0030430*	14	−1.3371	0.219871323	1
*hsa-miR-135a-5p*	*MIMAT0000428*	3	−1.3206	0.049728049	1
*hsa-miR-4507*	*MIMAT0019044*	14	−1.3176	0.284165056	1
*hsa-miR-16-1-3p*	*MIMAT0004489*	13	−1.3047	0.297303611	1
*hsa-miR-4732-5p*	*MIMAT0019855*	17	−1.3026	0.178680845	1
*hsa-miR-30c-2-3p*	*MIMAT0004550*	6	−1.2993	0.006492318	0.502959718
*hsa-miR-576-5p*	*MIMAT0003241*	4	−1.2962	0.036429558	1
*hsa-miR-3938*	*MIMAT0018353*	3	−1.2899	0.295549865	1
*hsa-miR-5589-3p*	*MIMAT0022298*	19	−1.2730	0.305462044	1
*hsa-miR-323a-3p*	*MIMAT0000755*	14	−1.2519	0.067319367	1
*hsa-miR-5189-5p*	*MIMAT0021120*	16	−1.1895	0.302501869	1
*hsa-miR-3133*	*MIMAT0014998*	2	−1.1883	0.346269763	1
*hsa-miR-9-3p*	*MIMAT0000442*	1	−1.1722	0.265080141	1
*hsa-miR-3178*	*MIMAT0015055*	16	−1.1722	0.265080141	1
*hsa-miR-494-3p*	*MIMAT0002816*	14	−1.1329	0.069170156	1
*hsa-miR-382-3p*	*MIMAT0022697*	14	−1.1112	0.186877213	1
*hsa-miR-19b-3p*	*MIMAT0000074*	13	−1.0940	0.073080409	1
*hsa-miR-4772-3p*	*MIMAT0019927*	2	−1.0569	0.343904631	1
*hsa-miR-5193*	*MIMAT0021124*	3	−1.0533	0.289583569	1
*hsa-miR-29c-3p*	*MIMAT0000681*	1	−1.0248	0.041438418	1
*hsa-miR-7152-5p*	*MIMAT0028214*	10	−1.0203	0.309763874	1
*hsa-miR-26a-5p*	*MIMAT0000082*	3	−1.0153	0.005038089	0.502959718

## Data Availability

The data presented in this study are available on request from the corresponding author.

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
