# Peer review of "Coronin 1C, Regulated by Multiple microRNAs, Facilitates Cancer Cell Aggressiveness in Pancreatic Ductal Adenocarcinoma"

_genes, 2023, doi:10.3390/genes14050995_

Round 1

Reviewer 1 Report

The manuscript entitled “Coronin 1C, regulated by multiple microRNAs, facilities cancer cell aggressiveness in pancreatic ductal adenocarcinoma” by Fukuda et al. reports for the first time the study of Coronin 1C in PDAC focusing their goals in verifying the involvement of Coronin 1C in PDAC malignancy by studying 5 putative microRNAs regulating Coronin 1C expression.

The aim of the work is interesting, but I suggest some point to review:

1.     Line 159, paragraph 3.2: explain the reason to study only Coronin 1C when it was stated that prognosis was significantly worse in patients with high expression of CORO1C and CORO2A versus low expression of these genes.

2.     Line 177: rephrase the sentence “however, we do not know the reason for this.”

3.     Pag. 7, Figure 3A: which is the difference between mock and control? Control is intended as “control siRNAs”? Please clarify this.

4.     Pag. 7, Figure 3A: I do not understand why mock condition fixed at 100% has standard deviation.

5.     Pag. 7, Figure 3A SW1990 cells: even if statistical analysis may result in a significance, the reduction of proliferation of SW1990 cells with siCORO1C-1 is too little to be considered as a reduction respect to mock condition. Which percentage is it?

6.     Pag. 11, Figure 6 lines 222-224: the sentence “CORO1C mRNA expression was reduced at 48 h after transfection of miR-26a-5p, miR-29c-3p, miR-148-5p, or miR-217 in PDAC cell lines (PANC-1 and SW1990). On the other hand, transfection with miR-130b-5p did not show remarkable regulation of CORO1C expression in PDAC cells.” is more suitable for the results section than for figure legend.

7.     Pag. 11, Figure 6 lines 227-228: the sentence “CORO1C protein expression according to Western blotting. CORO1C protein expression was reduced at 72 h after transfection of miR-26a-5p, miR-29c-3p, miR-130b-5p, miR-148-5p, or miR-217 in PDAC cells.” is more suitable for the results section than for figure legend.

8.     Pag. 11, Figure 6 lines 229-230: the sentence “Expression of CORO1C was reduced by miR-26a-5p, miR-29c-3p, miR-148-5p, or miR-217 transfection in PANC-1 and SW1990 cells.” is more suitable for the results section than for figure legend.

9.     I suggest to add densitometrical analysis of bands in figure 6B.

10. Paragraph 3.5: add description of results reported in Figures 7D and 7E and Figures 8D and 8E as they are not reported in the text.

11. Line 238: the authors stated that “cell proliferation was suppressed by miR-26a-5p transfection in PDAC cells (Figure 7A)” but the reduction is evident only in one cell line. Why? How can be justified this?

12. Did the authors try to do immunohistochemical analysis of CORO1C protein and microRNAs involved in regulation of CORO1C on slide tissues from pancreatic tumours to confirm the inverse relationship they have found?

Minor points:

1.     Line 173: correct Figure S2 with Figure S1 as only one supplementary figure is present.

2.     Pag. 10, Figure 5: the first panel should be named “A” and the second “B”. Adapt text and legend to this modification.

3.     Line 211: remove the dot in this part “GSE24279 (A).and”

4.     Line 285: I think it is better to put in italics “in vivo” and “in vitro”.

Author Response

Guest Editor

Dr. Manmeet Rawat

Dear Dr. Rawat,

We would like to express our gratitude for your consideration of our above-mentioned manuscript for publication in Genes. Enclosed, please find the revised manuscript (genes-2339379) along with a detailed explanation of the revisions, which were made based on the reviewers’ comments. All changes are highlighted in the revised manuscript.

Reviewer #1

The manuscript entitled “Coronin 1C, regulated by multiple microRNAs, facilities cancer cell aggressiveness in pancreatic ductal adenocarcinoma” by Fukuda et al. reports for the first time the study of Coronin 1C in PDAC focusing their goals in verifying the involvement of Coronin 1C in PDAC malignancy by studying 5 putative microRNAs regulating Coronin 1C expression.

The aim of the work is interesting, but I suggest some point to review:

Comment-1: Line 159, paragraph 3.2: explain the reason to study only Coronin 1C when it was stated that prognosis was significantly worse in patients with high expression of CORO1C and CORO2A versus low expression of these genes.

Response: Following the reviewer's comment, I added the following sentences (Results 3.2).

As results of the in silico analysis, two genes (CORO1C and CORO2A) were selected as candidate cancer-promoting genes that affect the prognosis of PDAC patients. Molecular analysis of these two genes is essential for elucidating the malignant transformation of PDAC cells. By comparing the significant difference in 5-year survival rate, this study focused on CORO1C and performed functional analysis.

Comment-2: Line 177: rephrase the sentence “however, we do not know the reason for this.”

Response: As suggested by the reviewer's comment, the text had modified as follows (Results 3.3).

The antitumor effects (cell proliferation and migration) of the two siRNAs, siCORO1C-1 and siCORO1C-2, differed (Figures 3A and 3B). Under the analysis conditions of this study, antitumor effects of siCORO1C-1 were insufficient compared to siCORO1C-2 in SW1990 cells.

Comment-3: Pag. 7, Figure 3A: which is the difference between mock and control? Control is intended as “control siRNAs”? Please clarify this.

Response: I apologize for the insufficient explanation of "mock" and "control". I added the explanation for mock and control as follows (Materials and Methods 2.4).

In this study, we define "mock" and "control" as follows; mock: transfection reagent was added to the medium, control: a scrambled RNA different from the siRNAs or miRNA sequence was added to the medium.

Comment-4: Pag. 7, Figure 3A: I do not understand why mock condition fixed at 100% has standard deviation.

Response: I apologize for my incorrect analysis of the graphs in Figure 3. Regarding Figure 3, we re-analyses again and present the corrected graphs in the revised version.

Comment-5: Pag. 7, Figure 3A SW1990 cells: even if statistical analysis may result in a significance, the reduction of proliferation of SW1990 cells with siCORO1C-1 is too little to be considered as a reduction respect to mock condition. Which percentage is it?

Response: In response to the reviewer’s suggestion, we had analysis Figure 3 again. I mentioned the anticancer effects of siCORO1C-1 as follows (Results 3.3).

The antitumor effects (cell proliferation and migration) of the two siRNAs, siCORO1C-1 and siCORO1C-2, differed (Figures 3A and 3B). Under the analysis conditions of this study, antitumor effects of siCORO1C-1 were insufficient compared to siCORO1C-2 in SW1990 cells.

Comment-6: Pag. 11, Figure 6 lines 222-224: the sentence “CORO1C mRNA expression was reduced at 48 h after transfection of miR-26a-5p, miR-29c-3p, miR-148-5p, or miR-217 in PDAC cell lines (PANC-1 and SW1990). On the other hand, transfection with miR-130b-5p did not show remarkable regulation of CORO1C expression in PDAC cells.” is more suitable for the results section than for figure legend.

Response: I agree with the reviewer's suggestion. Change the location of the description (Results 3.4).

Comment-7: Pag. 11, Figure 6 lines 227-228: the sentence “CORO1C protein expression according to Western blotting. CORO1C protein expression was reduced at 72 h after transfection of miR-26a-5p, miR-29c-3p, miR-130b-5p, miR-148-5p, or miR-217 in PDAC cells.” is more suitable for the results section than for figure legend.

Response: I agree with the reviewer's suggestion. Change the location of the description (Results 3.4).

Comment-8: Pag. 11, Figure 6 lines 229-230: the sentence “Expression of CORO1C was reduced by miR-26a-5p, miR-29c-3p, miR-148-5p, or miR-217 transfection in PANC-1 and SW1990 cells.” is more suitable for the results section than for figure legend.

Response: I agree with the reviewer's suggestion. Change the location of the description (Results 3.4).

Comment-9: I suggest to add densitometrical analysis of bands in figure 6B.

Response: I agree with the reviewer's suggestion. ImageJ was used to analyse the intensity of the bands for the Western blot in Figure 6B. The results of the ImageJ analyses are graphed and shown in Figure S2. The analysis results are described as follows (Results 3.4).

The intensity of the bands for the Western blotting were analysed by ImageJ software. The results of the ImageJ analyses were shown in Figure S3.

Comment-10: Paragraph 3.5: add description of results reported in Figures 7D and 7E and Figures 8D and 8E as they are not reported in the text.

Response: I apologize for the incomplete description. Added the following sentences (Results 3.5).

Typical images of Figures 7B (cell migration) and 7C (cell invasion) were shown in Figures 7D and 7E, respectively.

Typical images of Figures 8B (cell migration) and 8C (cell invasion) were shown in Figures 8D and 8E, respectively.

Comment-11: Line 238: the authors stated that “cell proliferation was suppressed by miR-26a-5p transfection in PDAC cells (Figure 7A)” but the reduction is evident only in one cell line. Why? How can be justified this?

Response: As suggested by the reviewer's comment, the text had modified as follows (Results 3.5).

The inhibitory effect of miR26a-5p on cell proliferation was observed in SW1990 cells, but not in PANC1 cells (Figure 7A).

Comment-12: Did the authors try to do immunohistochemical analysis of CORO1C protein and microRNAs involved in regulation of CORO1C on slide tissues from pancreatic tumours to confirm the inverse relationship they have found?

Response: Reviewer comment is an important issue for microRNA/cancer research. Investigating the localization of microRNAs in clinical tissues is an important theme, but the analysis method is extremely complicated. Clinical specimens need to be fixed for in situ analysis, but the tissues we have are paraffin blocks for pathological diagnosis. Therefore, the reviewer's suggestions (microRNA in situ and immunostaining) are not possible for this revised version.

In revised version, we investigated the correlations of the expression levels between five miRNAs (miR-26a-5p, miR-29c-3p, miR-130b-5p, miR-148a-5p and miR-217) and CORO1C by TCGA in silico analysis. The results of the analysis are shown in the Figure S3, and mentioned as follows (Results 3.4).

The correlations of expression levels between five miRNAs (miR-26a-5p, miR-29c-3p, miR-130b-5p, miR-148a-5p, and miR-217) and CORO1C were evaluated by using TCGA-PDAC data (Figure S2). A Spearman’s rank test indicated that negative correlation was detected in expression levels of miR-29a-3p and CORO1C in PDAC clinical specimens (p < 0.05, r = -0.3666). No inverse correlation was observed between the expression levels of other miRNAs and CORO1C.

Minor points:

Comment-1: Line 173: correct Figure S2 with Figure S1 as only one supplementary figure is present.

Response: I apologize for the incomplete numbering of the S-Figure. Corrected Figure numbers.

Comment-2: Pag. 10, Figure 5: the first panel should be named “A” and the second “B”. Adapt text and legend to this modification.

Response: As suggested by the reviewer's comment, the text had modified as follows (Results 3.4).

Five miRNAs (miR-26a-5p, miR-29c-3p, miR-130b-5p, miR-148a-5p, and miR-217) were commonly downregulated into two datasets. We confirmed that expression levels of five miRNAs were downregulated in PDAC tissues (n = 136) compared with normal tissues (n = 22) by using GSE 24279 data (Figure 5A). Analyzes using other dataset (GSE71533) also confirmed that the expression levels of five miRNAs was suppressed in PDAC tissues (n = 36) compared with normal tissues (n = 16) (Figure 5B).

Comment-3: Line 211: remove the dot in this part “GSE24279 (A).and”

Response: Corrected as pointed out by reviewer.

Comment-4: Line 285: I think it is better to put in italics “in vivo” and “in vitro”.

Response: Corrected as pointed out by reviewer.

Thank you for your constructive comments and suggestions. Some experiments could not be performed on the reviewer's comments. However, I have made some corrections as pointed out by the reviewers, and the quality is comparable to the papers that have been accepted in your journal “Genes” so far. Again, thank you for your consideration of our manuscript for publication in your journal.

Sincerely yours,

Naohiko Seki, Ph.D.

Department of Functional Genomics

Chiba University Graduate School of Medicine

1-8-1 Inohana, Chuo-ku,

Chiba 260-8670, Japan

Phone: +81-43-226-2971

Fax: +81-43-227-3442

Reviewer 2 Report

The authors investigated the functional correlation between CORO1C and PDAC, and identified the high expression of CORO1C in PDAC. CORO1C promoted the proliferation, migration, and invasion of cancer cells, and follow up studies showed that a set of miRNAs especially miR-26a-5p and miR-29c-3p suppress PDAC. Overall, the manuscript was organized and written well by providing sufficient background and updated references. This is an important study since PDAC is a highly aggressive cancer with an extremely low survival rate. However, some major flaws can be seen in the results section, which need careful clarification, discussion, and possible further experimental evidence.

1 Based on analysis, both CORO1C and CORO2A were highly expressed in PDAC and correlated with low survival, but why you just focused on CORO1C study? Are there any publications reporting the effect of CORO2A on PDAC or other types of cancers?

2 In lines 163-164, please add references for your statement.

3 In line 173, “Figure S2” should be “Figure S1”.

4 Several concerns were found in Figure 3. First, what was the difference between “mock” and “control” (also for Figures 6, 7, 8, S1)? If control stands for non-targeting siRNA, the statistical analysis should be made for control vs siCORO1C, rather than mock vs siCORO1C. Second, how did you normalize your data? In panel 3A, it seems like you normalized mock to 100, but in panels 3B and 3C, they were confusingly shown that none of four conditions were normalized to 100. Third, SW1990 data in 3B was exactly same as that of 3A. Please double check the figure usage and get rid of this potential duplication.

5 In lines 192-193, please explain how you proposed your hypothesis, namely, why you hypothesized that miRNA could negatively regulate CORO1C expression?

6 In Figure 6A, it looks like the “control” consistently decreased CORO1C mRNA level. Was control the transfected empty vector? Please explain the give possible reasons.

7 In Figure 6B, the change of CORO1C in SW1990 cells were not obvious. You may calculate the intensity of western blot bands.

8 In lines 237-238, the interpretation was not quite reasonable, since no change can be found in PANC-1 cells.

Author Response

Guest Editor

Dr. Manmeet Rawat

Dear Dr. Rawat,

We would like to express our gratitude for your consideration of our above-mentioned manuscript for publication in Genes. Enclosed, please find the revised manuscript (genes-2339379) along with a detailed explanation of the revisions, which were made based on the reviewers’ comments. All changes are highlighted in the revised manuscript.

Reviewer #2

The authors investigated the functional correlation between CORO1C and PDAC, and identified the high expression of CORO1C in PDAC. CORO1C promoted the proliferation, migration, and invasion of cancer cells, and follow up studies showed that a set of miRNAs especially miR-26a-5p and miR-29c-3p suppress PDAC. Overall, the manuscript was organized and written well by providing sufficient background and updated references. This is an important study since PDAC is a highly aggressive cancer with an extremely low survival rate. However, some major flaws can be seen in the results section, which need careful clarification, discussion, and possible further experimental evidence.

Comment-1: Based on analysis, both CORO1C and CORO2A were highly expressed in PDAC and correlated with low survival, but why you just focused on CORO1C study? Are there any publications reporting the effect of CORO2A on PDAC or other types of cancers?

Response: Following the reviewer's comment, I added the following sentences (Results 3.2). As for CORO2A genes, we are currently analyzing the functional significance of pancreatic cancer, so we will omit it in this paper. I appreciate your understanding.

As results of the in silico analysis, two genes (CORO1C and CORO2A) were selected as candidate cancer-promoting genes that affect the prognosis of PDAC patients. Molecular analysis of these two genes is essential for elucidating the malignant transformation of PDAC cells. By comparing the significant difference in 5-year survival rate, this study focused on CORO1C and performed functional analysis.

Comment-2: In lines 163-164, please add references for your statement.

Response: As suggested by the reviewer’s comment, Added the following sentences (Discussion 2nd chapter).

Furthermore, we investigated the CORO1C-mediated molecular pathways in PDAC. GSEA analysis revealed that “epithelial–mesenchymal transition (EMT)” was the most enriched pathway in the CORO1C high expression group. EMT has widely been accepted as a critical step in metastatic dissemination in PDAC cells, and various functional RNAs are closely involved in this process [39,40]. However, EMT-independent pathways have also been reported to be involved in distant metastases of PDAC [41]. Exploring the molecular pathways mediated by CORO1C will provide hints for elucidating the invasion and metastasis mechanisms of PDAC cells.

Comment-3: In line 173, “Figure S2” should be “Figure S1”.

Response: I apologize for the incomplete numbering of the S-Figure. Corrected Figure numbers.

Comment-4: Several concerns were found in Figure 3. First, what was the difference between “mock” and “control” (also for Figures 6, 7, 8, S1)? If control stands for non-targeting siRNA, the statistical analysis should be made for control vs siCORO1C, rather than mock vs siCORO1C. Second, how did you normalize your data? In panel 3A, it seems like you normalized mock to 100, but in panels 3B and 3C, they were confusingly shown that none of four conditions were normalized to 100. Third, SW1990 data in 3B was exactly same as that of 3A. Please double check the figure usage and get rid of this potential duplication.

Response: I apologize for my incorrect analysis of the graphs in Figure 3. Regarding Figure 3, we re-analyses again and present the corrected graphs in the revised version.

From our experiments so far, we have often experienced that the expression levels of various genes altered when control-siRNAs and control-miRNAs are transfected into cells. It is difficult to completely control this phenomenon. Therefore, verification of changes due to target siRNAs and miRNAs to be tested should be compared with mock.

Comment-5: In lines 192-193, please explain how you proposed your hypothesis, namely, why you hypothesized that miRNA could negatively regulate CORO1C expression?

Response: As suggested by the reviewer’s comment, I added the following sentences (Results 3.4).

A vast number of studies have revealed that gene expression is regulated by various functional RNAs, e.g., lncRNAs, miRNAs, and circRNAs. Among them, it has been clarified that dysregulation of miRNAs causes aberrant expression of cancer-related genes, and is closely involved in malignant transformation of cancer cells.

Comment-6: In Figure 6A, it looks like the “control” consistently decreased CORO1C mRNA level. Was control the transfected empty vector? Please explain the give possible reasons.

Response: I apologize for the insufficient explanation of "mock" and "control". I added the explanation for mock and control as follows (Materials and Methods 2.4).

In this study, we define "mock" and "control" as follows; mock: transfection reagent was added to the medium, control: a scrambled RNA different from the siRNAs or miRNA sequence was added to the medium.

From our experiments so far, we have often experienced that the expression levels of various genes altered when control-siRNAs and control-miRNAs are transfected into cells. It is difficult to completely control this phenomenon. Therefore, verification of changes due to target siRNAs and miRNAs to be tested should be compared with mock.

Comment-7: In Figure 6B, the change of CORO1C in SW1990 cells were not obvious. You may calculate the intensity of western blot bands.

Response: I agree with the reviewer's suggestion. ImageJ was used to analyse the intensity of the bands for the Western blot in Figure 6B. The results of the ImageJ analyses are graphed and shown in Figure S3. The analysis results are described as follows (Results 3.4).

The intensity of the bands for the Western blotting were analysed by ImageJ software. The results of the ImageJ analyses were shown in Figure S3.

Comment-8: In lines 237-238, the interpretation was not quite reasonable, since no change can be found in PANC-1 cells.

Response: As suggested by the reviewer's comment, the text had modified as follows (Results 3.5).

The inhibitory effect of miR-26a-5p on cell proliferation was observed in SW1990 cells, but not in PANC-1 cells (Figure 7A).

Thank you for your constructive comments and suggestions. Some experiments could not be performed on the reviewer's comments. However, I have made some corrections as pointed out by the reviewers, and the quality is comparable to the papers that have been accepted in your journal “Genes” so far. Again, thank you for your consideration of our manuscript for publication in your journal.

Sincerely yours,

Naohiko Seki, Ph.D.

Department of Functional Genomics

Chiba University Graduate School of Medicine

1-8-1 Inohana, Chuo-ku,

Chiba 260-8670, Japan

Phone: +81-43-226-2971

Fax: +81-43-227-3442

Round 2

Reviewer 2 Report

The authors have submitted a revised manuscript and responses to comments, and I appreciate that all my concerns have been addressed with substantial ameliorations. There is one more point which needs to be considered: the new Fig 1 has overlapping in which the panel A is totally covered by panel B. Please resubmit a new version of manuscript with updated Fig 1.

Author Response

Guest Editor

Dr. Manmeet Rawat

Dear Dr. Rawat,

We would like to express our gratitude for your consideration of our above-mentioned manuscript for publication in Genes.

Reviewer #2 (Minor Revisions)

Comment: The authors have submitted a revised manuscript and responses to comments, and I appreciate that all my concerns have been addressed with substantial ameliorations. There is one more point which needs to be considered: the new Fig 1 has overlapping in which the panel A is totally covered by panel B. Please resubmit a new version of manuscript with updated Fig 1.

Response: I apologize for the clumsiness in uploading the paper. I will upload the thesis again, so please check it.

Thank you for your constructive comments and suggestions. Again, thank you for your consideration of our manuscript for publication in your journal.

Sincerely yours,

Naohiko Seki, Ph.D.

Department of Functional Genomics

Chiba University Graduate School of Medicine

1-8-1 Inohana, Chuo-ku,

Chiba 260-8670, Japan

Phone: +81-43-226-2971

Fax: +81-43-227-3442
